# Deep learning-based prediction of kinetic parameters from myocardial perfusion MRI

**Cian M. Scannell**[*1,2]               CIAN.SCANNELL@KCL.AC.UK
**Piet van den Bosch**[*3]            P.C.J.V.D.BOSCH@STUDENT.TUE.NL
**Amedeo Chiribiri**[1]             AMEDEO.CHIRIBIRI@KCL.AC.UK
**Jack Lee**[1]                   JACK.LEE@KCL.AC.UK
**Marcel Breeuwer**[3,4]           MARCEL.BREEUWER@PHILIPS.COM
**Mitko Veta**[3]                  M.VETA@TUE.NL

[1] *School of Biomedical Engineering and Imaging Sciences, King's College London, United Kingdom*

[2] *The Alan Turing Institute London, United Kingdom*

[3] *Department of Biomedical Engineering, Medical Image Analysis group, Eindhoven University of Technology, Eindhoven, The Netherlands*

[4] *Philips Healthcare, Best, The Netherlands*

## 1. Introduction

Quantification of myocardial perfusion, using tracer-kinetic modelling, from dynamic contrast-enhanced (DCE-) magnetic resonance imaging (MRI) provides an automated and user-independent alternative to the visual assessment of the images with a high prognostic value (Sammut et al., 2017). With tracer-kinetic modelling, the passage of contrast agent from the left ventricle (LV) to the myocardium is modelled as a system with two interacting compartments - the plasma and the interstitium. This model gives a pair of coupled differential equations which describe the evolution of the contrast agent as a non-linear function of physiological parameters, such as myocardial blood flow (MBF). Given the concentration of contrast agent in the LV (the arterial input function (AIF)) and the myocardium observed from the DCE-MRI examination, we can then solve the inverse problem in order to estimate the kinetic parameters. This parameter estimation problem is traditionally solved using a non-linear least squares fitting algorithm.

Recent work suggests that the use of Bayesian inference provides a more accurate and reproducible estimation of the kinetic parameters than the least-squares fitting approach (Dikaios et al., 2017). However, a major drawback of this approach is the time complexity of the Markov chain Monte Carlo (MCMC) sampling required in order to accurately approximate the posterior distribution of the parameters.

In this work, a convolutional neural network (CNN) is used to directly predict MBF based on the DCE-MRI data. The network processes the data in a voxel-wise manner. It takes both the AIF and myocardial tissue curve as input and directly predicts the MBF value without the explicit use of the tracer-kinetic model. The regression problem is trained using target values as obtained from the Bayesian inference. This leads to the construction of a

---

[*] Contributed equally

model with similar performance to the Bayesian inference but with reduced computation cost.

## 2. Methods

The dataset consisted of nine patients (five healthy, four diseased), giving 27893 distinct myocardial voxels. The target MBF values were computed using a previously developed Bayesian inference scheme (Scannell et al., 2019). The network architecture used in this work is based on that of Ho et al. (Ho et al., 2016). The input to the network for a given voxel is the AIF and the three-by-three neighbourhood of tissue curves centred at the voxel of interest. The neighbourhood of tissue curves is used as a spatial regularisation to enforce the spatial smoothness of the predicted MBF maps. The nine curves are processed as nine input feature channels. The AIF and tissue curves are initially processed separately, using convolutional layers, in two distinct branches of the network. This acts as a feature extractor on the curves. These features are then concatenated and used as the input to a series of fully-connected layers which leads to a prediction of MBF (Figure 1(a)).

Both branches of the network separately consist of a series of convolutional layers, each followed by a max pooling layer. Every trainable layer was followed by a ReLU activation. The model used the Adam optimisation algorithm with initial learning rate of 0.0005 which optimised the mean squared error (MSE) between the predicted and target MBF values. Early stopping was employed with a patience of 40 epochs.

Due to the small number of patient datasets available, cross-validation was used to evaluate the performance of the model. The nine patient datasets were randomly divided into five training sets, two validation sets, and two test sets in 10 different ways. This was done randomly with the constraint that each patient appears in the test set at least once and the average of the test performance on each of the 10 test sets is reported. The diagnostic accuracy was assessed through a blinded comparison of the clinical interpretation of the MR images and the clinical interpretation of, solely, the MBF maps.

## 3. Results

The median MBF value (25th percentile, 75th percentile) obtained from the Bayesian inference was 2.35 (1.9, 2.68) mL/min/mL. The mean (standard deviation) MSE between the predicted and target MBF values was 0.096 (0.081) mL/min/mL. This corresponds to an average voxel error in MBF of 4% when using the deep learning-based prediction. The assessment of the patients' health based purely on the MBF maps obtained using the deep learning-based prediction matches the clinical report in all slices. A comparison of the MBF maps obtained from the Bayesian inference and the deep learning prediction is shown in Figure 1(b). The use of the CNN-based MBF predictions reduced the processing time for a representative slice from 28 minutes (Bayesian inference) to two seconds.

## 4. Discussion and Conclusion

The small MSE reported indicates that the deep learning approach can well approximate the MBF values computed using Bayesian inference without the need for the time intensive

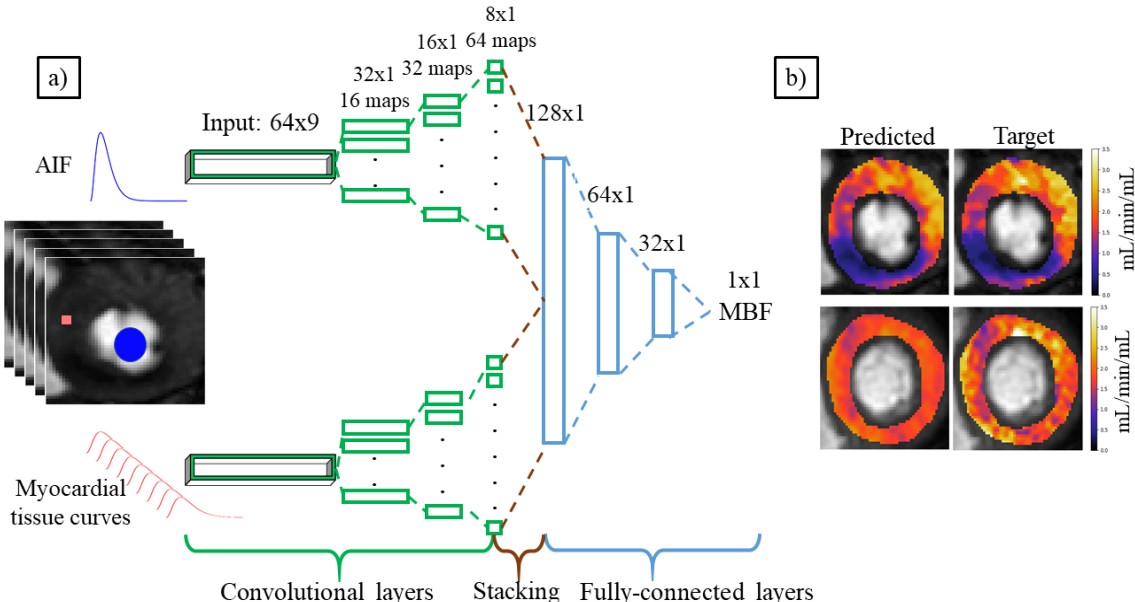

Figure 1: (a) The architecture which consists of two distinct convolutional branches which process the AIF and myocardial curves independently. The learned features are then concatenated and passed through a series of fully-connected layers to predict the MBF. (b) A comparison of the predicted and target MBF maps for two patients, one with a clear perfusion defect (top) and one healthy (bottom).

MCMC sampling. The deep learning model can process an imaging slice in the order of seconds in contrast to the Bayesian inference which can take up to 30 minutes per slice. Crucially, the deep learning model provides a similar level of diagnostic information, as evidenced by the diagnostic accuracy. However, as seen in Figure 1(b), the MBF maps do look, in comparison, rather simplified or smoothed. It seems that the model was unable to learn more complex features to capture the finer detail in the curves. This could potentially be improved with the use of a larger training set. It is also possible that there is excessive spatial regularisation being caused by the use of the neighbouring voxels as input to the model.

We experimented with training the model using simulated data, where the ground-truth parameter values are known, but this model was not able to generalise well to the patient data. Fine-tuning the simulation model on the patient data did not provide any advantages over training solely on the patient data. However, it may be possible to improve these results by making the simulations more realistic with respect to the patient data.

A further advantage to the deep learning approach is that there is a time delay between the AIF and tissue curves which is typically estimated as an additional model parameter. However, the CNN was trained directly on the observed curves without accounting for this delay. Due to nature of the convolutional layers, the network was able to learn an invariance to this time delay, leading to one less source of error in the estimation.

## Acknowledgments

The authors acknowledge financial support from the King's College London & Imperial College London EPSRC Centre for Doctoral Training in Medical Imaging (EP/L015226/1); Philips Healthcare; The Department of Health via the National Institute for Health Research (NIHR) comprehensive Biomedical Research Centre award to Guy's & St Thomas' NHS Foundation Trust in partnership with King's College London and King's College Hospital NHS Foundation Trust; The Centre of Excellence in Medical Engineering funded by the Wellcome Trust and EPSRC under grant number WT 088641/Z/09/Z; The Alan Turing Institute (EPSRC grant no. EP/N510129/1).

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
