# OpenReview forum: "Deep learning-based prediction of kinetic parameters from myocardial perfusion MRI"
_MIDL.io/2019/Conference/Abstract — MIDL Abstract 2019_

### Official Review · AnonReviewer2 · 2019-04-29
**Interesting work but rather small dataset**

**Rating:** 3
**Confidence:** 3

**Review:**

The authors propose an approach to do model free estimation of myocardial perfusion using convolutional neural network trained to mimic and produce similar results as the current state of the art: bayesian estimation. The results seem convincing, both in terms of qualitative and quantitative evaluation. The dataset that has been used in this work is rather small and decreases my confidence in these results. On the other hand, the authors have done a 10 fold cross validation for the approach, which kind of reduces my lack of confidence.

The input to the network and architecture is not really clear to me. I have understood that the architecture is multi-branch and that the types of input used here are rather heterogeneous, but a couple of crucial things are still unclear. Is the prediction voxel-wise? what kind of image content is presented as input? how was the data normalized?

In my opinion a brief discussion about these aspects will be beneficial for the paper.

Apart from the method section, the approach has been clearly detailed and explained (given the length constraints imposed by the abstract format). The discussion is appropriate and sound.

---

### Official Review · AnonReviewer1 · 2019-05-01
**even too few details on the actual proposed approach, but the solution may be promising for nice speed up.**

**Rating:** 3
**Confidence:** 2

**Review:**

The abstract describe well on the motivation of the problem to solve, but too few descriptions on the technical approach. The reason for arguing "acceptance" is mainly for the new approach to replace MCMC with a good potential of computational efficiency.

---

### Decision · Program_Chairs · 2019-05-06
**Acceptance Decision**

Accept